 Select                                    SciPost Phys. 8, 067 (2020)

# Operator entanglement in local quantum circuits I: Chaotic dual-unitary circuits

**Bruno Bertini⋆, Pavel Kos and Tomaz Prosen**

Department of physics, FMF, University of Ljubljana,
Jadranska 19, SI-1000 Ljubljana, Slovenia

⋆ bruno.bertini@fmf.uni-lj.si

## Abstract

The entanglement in operator space is a well established measure for the complexity of quantum many-body dynamics. In particular, that of local operators has recently been proposed as *dynamical chaos indicator*, i.e. as a quantity able to discriminate between quantum systems with integrable and chaotic dynamics. For chaotic systems the local-operator entanglement is expected to grow linearly in time, while it is expected to grow at most logarithmically in the integrable case. Here we study the dynamics of local-operator entanglement in dual-unitary quantum circuits, a class of "statistically solvable" quantum circuits that we recently introduced. We identify a class of "completely chaotic" dual-unitary circuits where the local-operator entanglement grows linearly and we provide a conjecture for its asymptotic behaviour which is in excellent agreement with the numerical results. Interestingly, our conjecture also predicts a "phase transition" in the slope of the local-operator entanglement when varying the parameters of the circuits.

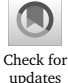

# 1   Introduction

The complexity of classical dynamical systems is customarily characterised by the Kolmogorov-Sinai entropy, which quantifies the rate of information-entropy produced by the dynamics. This quantity is related to the Lyapunov exponents via the Pesin formula [1–4] and can be thought of as a measure for the sensitivity of the dynamics to the initial conditions — the famous "butterfly effect". Such an appealing connection continues to work for quantum systems in the semiclassical regime, or, more generally, for quantum systems with a large parameter $N$ identifying a small effective Planck constant $\hbar_{\text{eff}} = 1/N$. In these cases the complexity is efficiently quantified by out-of-time-ordered correlation functions [5–25]. The "quantum Lyapunov exponents" obtained in this way, however, make sense only up to times of the order $\log N \sim \log(1/\hbar_{\text{eff}})$, hence, such a quantum dynamical chaos can only be justified in the semiclassical limit $\hbar_{\text{eff}} \to 0$: the quantification of dynamical chaos in quantum many-body systems with $\hbar_{\text{eff}} \sim 1$, such as quantum spin-1/2 chains, turned out to be much harder [26]. An appealing algebraic generalisation of Kolmogorov-Sinai entropy to (non-commutative) quantum dynamical systems exists due to Alicki and Fannes [27], but it cannot discriminate between integrable and non-integrable dynamics: the Alicki-Fannes dynamical entropy is typically positive even for non-interacting (quasi-free) extended systems [26].

A different route for measuring quantum dynamical chaos can be found by considering the algorithmic complexity of state-of-the-art classical simulations of the quantum dynamics, say, using matrix-product-state methods [28]. In particular, a good indicator of quantum dynamical complexity has been identified by looking at the Heisenberg evolution of operators that are initially localised in a small portion of the real space [29]. The idea is to think of a time-dependent operator as a state in an appropriate Hilbert space (the operator-space) and characterise the complexity of its evolution using the entanglement of such a state [30, 31]. This quantity, termed *operator-space entanglement*, or simply *operator entanglement*, seems to characterise very effectively the genuine dynamical complexity of quantum many-body systems [30–37]. Note that here we consider the operator entanglement dynamics for operators initially supported on a small spacial region, while related concepts have been considered for some manifestly non-local objects as well, such as, e.g., the time-dependent many-body propagator [38–42]. For this reason, in this work we will always refer to this quantity as *local-operator entanglement*.

The main problem is that, to date, there are essentially no exact benchmarks for the dynam-

ics of the local-operator entanglement except for certain results in integrable models [30–33] and in random models in a particular asymptotic limit [37]. Providing such exact benchmarks for *non-integrable* many-body quantum dynamical systems is our main objective. We consider systems represented as local quantum circuits [43–55], i.e. qudit chains (quantum spin-$(d-1)/2$ chains with arbitrary integer $d \geq 2$) where the time evolution is generated by the discrete application of unitary operators coupling neighbouring sites. We measure the local-operator entanglement through Rényi entropies at integer Rényi order. Our strategy is to write them in terms of partition functions on a non-trivial space-time domain that we contract in terms of row and corner transfer matrices. We divide this endeavour into two separate works investigating two conceptually distinct classes of quantum circuits that are generically non-integrable. Using the special properties of these classes we prove and conjecture some exact statements about the dynamics of the local-operator entanglement.

In the present paper, we study the dynamics of the local-operator entanglement for *dual-unitary* local quantum circuits. This is a class of local quantum circuits where the dynamics remains unitary also when the roles of space and time are swapped [56]. As we showed in a recent series of works, dual-unitarity is an extremely powerful property and enables the exact calculation of many statistical and dynamical properties. These include spectral statistics [57], (state) entanglement spreading [58], and dynamical correlations [56] (see also [54] and [59] for other useful features of dual-unitarity). Focussing on dual-unitary quantum circuits with *no local conservation laws* — the *chaotic* subclass — we conjecture a general formula for the dynamics of the local-operator entanglement. The idea is to compute the local-operator entanglement by considering separately the entanglement produced by the two edges of the spreading operator — as if the opposite edge were effectively sent to infinity. Dual-unitarity allows us to evaluate these contributions exactly revealing a simple and remarkable prediction, which is in excellent agreement with exact short-time numerical results. First, we find that in chaotic dual-unitary circuits the local-operator entanglement always grows linearly with time. Second, the slope of growth displays an abrupt transition when varying the parameters of the circuits. Third, the slope is maximal on one side of the transition. This has to be contrasted with the linear growth observed in Haar-random noisy circuits [37], where the slope is around half of the maximal one. These results once again put forward dual-unitary circuits (in appropriate parameter ranges) as minimal models — with fixed local Hilbert space dimension and local interactions — for the maximally-chaotic dynamics.

In the companion paper [60] (Paper II) we consider the dynamics of local-operator entanglement in local quantum circuits exhibiting local dynamical conservation laws, i.e. solitons. These conservation laws are generically not enough to generate an integrable structure à la Yang-Baxter. Limiting to the circuits of qubits ($d = 2$), we classify all instances of circuits with solitons and show that if a spreading operator crosses some soliton, the dynamics of its local-operator entanglement can be computed explicitly and exhibits saturation. Interestingly, we show that all circuits admitting moving solitons are dual-unitary. Importantly, since they have conservation laws, those dual-unitary circuits are *not chaotic* as the ones studied here.

The rest of this paper is laid out as follows. In Section 2 we give a detailed definition of the quantum many-body systems of interest for this work — local quantum circuits — and introduce a useful diagrammatic representation to study their dynamics. In Section 3 we introduce the local operator entanglement entropies and write them in terms of partition functions on appropriate space-time surfaces. In Section 4 we specialise the treatment to dual-unitary local quantum circuits, recalling their main defining features and characterising the "completely chaotic" class of interest in this paper. In Section 5 we formulate our conjecture and use it to explicitly compute the local-operator entanglement dynamics (explicitly comparing it with the numerical evaluation of the space-time partition functions). Finally, Section 6 contains our conclusions. Five appendices complement the main text with a number of minor technical

points.

## 2  Local Quantum Circuits

In this work we consider periodically-driven quantum many-body systems represented as *local quantum circuits*. These systems consist of a periodic chain of $2L$ sites, where at each site is embedded a $d$-dimensional local Hilbert space $\mathcal{H}_1 = \mathbb{C}^d$ so that the total Hilbert space is

$$\mathcal{H}_{2L} = \mathcal{H}_1^{\otimes 2L}. \tag{1}$$

The time evolution in the system is discrete and each time-step is divided into two halves. In the first half the time evolving operator is

$$\mathbb{U}^{\mathrm{e}} = \bigotimes_{x \in \mathbb{Z}_L} U_{x,x+1/2}, \tag{2}$$

where $\mathbb{Z}_L = \mathbb{Z} \cap (-L/2, L/2]$, while $U_{x,y} \in U(\mathcal{H}_1 \otimes \mathcal{H}_1)$ is the unitary "local gate" connecting the qdits at sites $x$ and $y$ and encoding all physical properties of a given quantum circuit. In the second half, instead, the system is evolved by

$$\mathbb{U}^{\mathrm{o}} = \mathbb{T}_{2L} \mathbb{U}^{\mathrm{e}} \mathbb{T}_{2L}^{\dagger}, \tag{3}$$

where $\mathbb{T}_{\ell}$ is a $\ell$-periodic translation by one site

$$\mathbb{T}_{\ell} \left( o_1 \otimes o_2 \cdots \otimes o_{\ell} \right) \mathbb{T}_{\ell}^{\dagger} \equiv o_2 \otimes o_3 \cdots \otimes o_{\ell} \otimes o_1. \tag{4}$$

Here $\{o_j\}$ are generic operators in $\mathcal{H}_1$. In summary, the "Floquet operator" — the time evolution operator for one period of the drive (one time-step) — is given by

$$\mathbb{U} = \mathbb{U}^{\mathrm{o}} \mathbb{U}^{\mathrm{e}} = \mathbb{T}_{2L} \mathbb{U}^{\mathrm{e}} \mathbb{T}_{2L}^{\dagger} \mathbb{U}^{\mathrm{e}}. \tag{5}$$

Note that, since the local gate is unitary, the Floquet operator $\mathbb{U}$ is also unitary. Moreover, from the definition (5) it immediately follows that $\mathbb{U}$ is invariant under two-site shifts

$$\mathbb{U} \mathbb{T}_{2L}^2 = \mathbb{T}_{2L}^2 \mathbb{U}. \tag{6}$$

Note that in this work we consider translationally invariant quantum circuits which are specified by a single 2-qudit gate $U_{x,x+1/2} = U$ for all $x \in \frac{1}{2}\mathbb{Z}_{2L}$, while we expect that the formalism we develop here should be useful also for generalizations to disordered and/or noisy quantum circuits.

Local quantum circuits admit a convenient diagrammatic representation. One depicts states as boxes with $2L$ outgoing legs (or wires) representing the local sites and operators as boxes with a number of incoming and outgoing legs corresponding to the number of local sites they act on. Each leg carries a Hilbert space $\mathcal{H}_1$. For instance, the identity operator on a single site, $\mathbb{1}$, is represented as

$$\mathbb{1} = \Big| \ , \tag{7}$$

while a generic single-site operator $a$ is represented as

$$a = \begin{array}{c}\bullet\end{array}. \tag{8}$$

The local gate and its Hermitian conjugate are instead represented as

$$U = \ \text{(red box)} \ , \qquad\qquad U^{\dagger} = \ \text{(blue box)} \ , \tag{9}$$

where we added a mark ¬ to stress that $U$ and $U^\dagger$ are generically not symmetric under space reflection (left to right flip) and time reversal (up to down flip, transposition of $U$). The time direction runs from bottom to top, hence lower legs correspond to incoming indices (matrix row) and upper legs to outgoing indices (matrix column). With these conventions, the diagrammatic representation of $\mathbb{U}$ reads as

$$\mathbb{U} = \quad , \tag{10}$$

where we labelled sites $x$ by half integers, $x \in \frac{1}{2}\mathbb{Z}_{2L}$, and boundary conditions in space (horizontal direction) are periodic. This means that the ultralocal operator

$$a_y \equiv \underbrace{\mathbb{1} \otimes \cdots \otimes \mathbb{1}}_{L-1+2y} \otimes\, a \otimes \underbrace{\mathbb{1} \otimes \cdots \otimes \mathbb{1}}_{L-2y}, \tag{11}$$

evolved up to time $t$ is represented as

$$a_y(t) \equiv (\mathbb{U}^\dagger)^t a_y \mathbb{U}^t = \qquad . \tag{12}$$

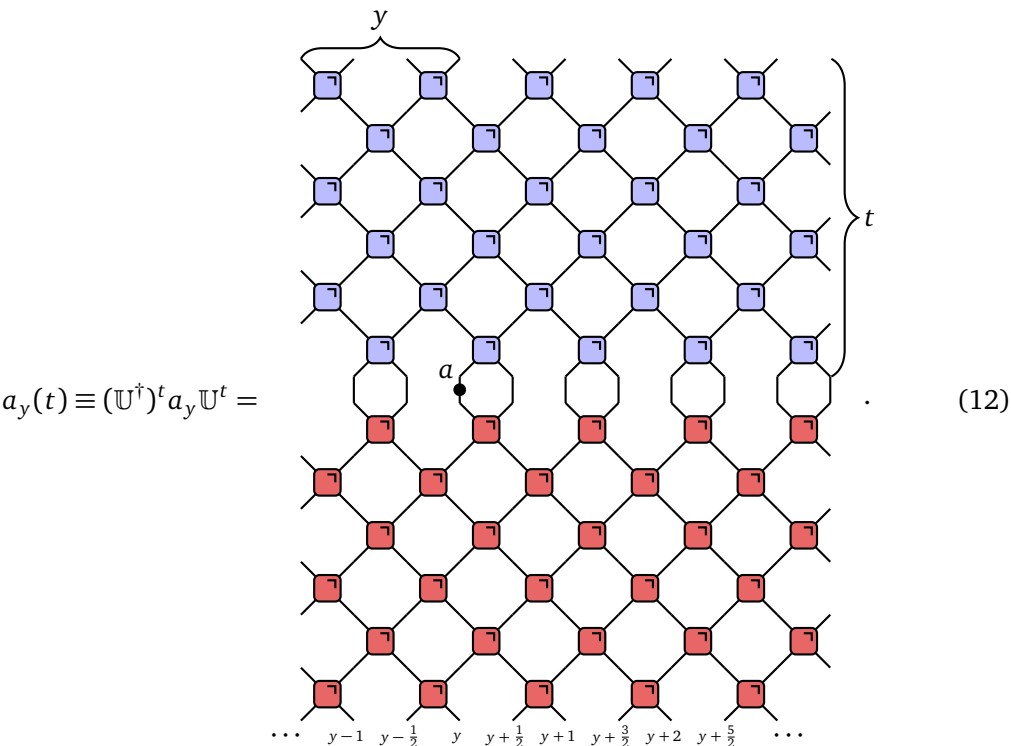

Before concluding we note that time-evolving operators transform covariantly under the following *gauge transformation* in the space of local gates

$$U \mapsto (u \otimes v) U (v^\dagger \otimes u^\dagger), \qquad u, v \in \mathrm{U}(d). \tag{13}$$

Specifically, we have

$$a_y(t) \equiv (\mathbb{U}^\dagger)^t a_y \mathbb{U}^t \mapsto \begin{cases} (v \otimes u)^{\otimes L} (\mathbb{U}^\dagger)^t (u^\dagger a u)_y \mathbb{U}^t (v^\dagger \otimes u^\dagger)^{\otimes L} & y \in \mathbb{Z}_L + \frac{1}{2} \\ (v \otimes u)^{\otimes L} (\mathbb{U}^\dagger)^t (v^\dagger a v)_y \mathbb{U}^t (v^\dagger \otimes u^\dagger)^{\otimes L} & y \in \mathbb{Z}_L \end{cases}. \tag{14}$$

## 2.1 Operator-to-state mapping

The time evolution of operators in $\mathcal{H}$ can be mapped into that of states in the "doubled" Hilbert space $\mathcal{H} \otimes \mathcal{H}$ by performing an operator-to-state (or vectorization) mapping

$$\text{End}(\mathcal{H}) \longrightarrow \mathcal{H} \otimes \mathcal{H}. \tag{15}$$

Choosing *any* basis $\{|n\rangle\}$ of $\mathcal{H}$ we completely specify the mapping by defining

$$|m\rangle \langle n| \quad \longmapsto \quad |m\rangle \otimes |n\rangle^*, \tag{16}$$

so that the time evolution maps to

$$a_y(t) \quad \longmapsto \quad |a_y(t)\rangle \equiv \sum_{n,m} \langle n|a_y(t)|m\rangle |n\rangle \otimes |m\rangle^* = (\mathbb{U}^\dagger \otimes \mathbb{U}^{\dagger *})^t |a_y\rangle. \tag{17}$$

The complex conjugation $(\cdot)^*$ is defined such that

$$^*\langle n|O^*|m\rangle^* = (\langle n|O|m\rangle)^*, \tag{18}$$

meaning that the vectorization mapping is linear (and not antilinear!) with respect to both, the ket and the bra parts[1].

For convenience we arrange the states $|n\rangle \otimes |m\rangle^*$ in $\mathcal{H} \otimes \mathcal{H}$ in such a way that the time evolution generated by $\mathbb{U}^\dagger \otimes \mathbb{U}^{\dagger *}$ is "local in space". Specifically,

$$|i_1 \dots i_{2L}\rangle \otimes |j_1 \dots j_{2L}\rangle^* = |i_1 j_1\rangle \otimes \cdots |i_{2L} j_{2L}\rangle, \tag{19}$$

where $\{|i\rangle; i = 1, 2, \dots, d\}$ is a real, orthonormal basis of $\mathcal{H}_1$. In general, for any set of states $|a\rangle, |b\rangle \cdots \in \mathcal{H}_1$, we use a compact notation $|a b \dots\rangle = |a\rangle \otimes |b\rangle \otimes \cdots$.

The mapping defined in this way is directly represented by folding the circuit

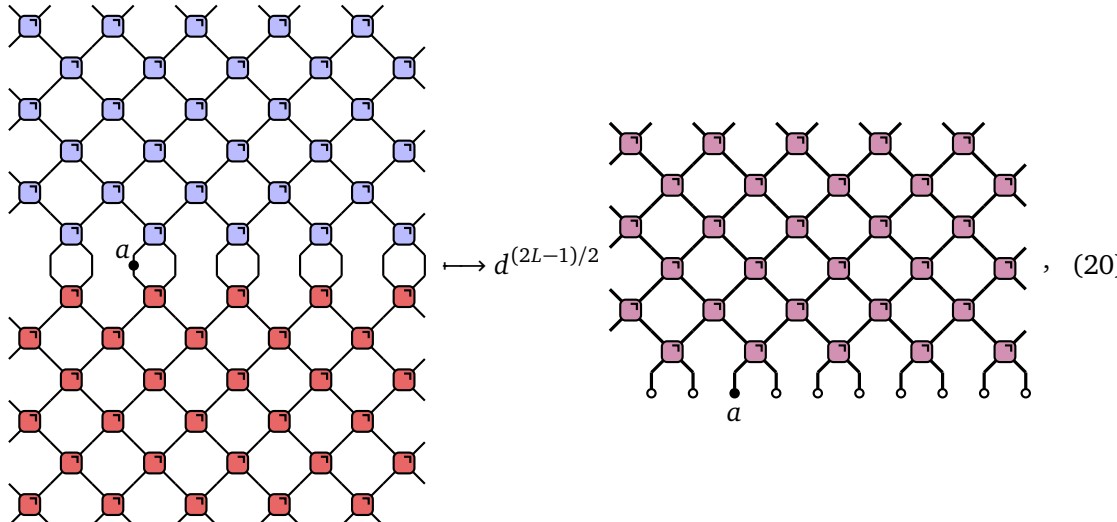

$$\tag{20}$$

where each thick wire carries a $d^2$ dimensional local Hilbert space $\mathcal{H}_1 \otimes \mathcal{H}_1$

$$\tag{21}$$

---

[1]We can always decide to choose a fixed canonical basis such that $|n\rangle^* = |n\rangle$.

and we introduced the "double gate"

$$W = \quad = \quad . \tag{22}$$

Note that the red gate is upside down, meaning that $U$ is transposed (c.f. $(\mathbb{U}^\dagger)^* = \mathbb{U}^T$ on the r.h.s. of (17)). Finally, we also introduced the (normalised) local states associated to to the identity operator

$$\frac{1}{\sqrt{d}} \; \Big| \; \longmapsto \; \frac{1}{\sqrt{d}} \; \bigcup \; = \; \equiv | \circ \rangle \,, \tag{23}$$

and to the operator $a$

$$a \; \longmapsto \; \bigcup_a \; = \; \Big|_a \; \equiv | a \rangle \,. \tag{24}$$

We stress that in this paper we always consider local operators that are Hilbert-Schmidt normalised

$$\text{tr}[aa^\dagger] = 1 \,. \tag{25}$$

For non-normalised operators one should include the appropriate normalisation factors in (20) and (24).

Finally, we remark that from the unitarity of $U$ it follows

$$\quad = \quad , \qquad \quad = \quad , \tag{26a}$$

$$\quad = \quad , \qquad \quad = \quad , \tag{26b}$$

where we introduced

$$\quad = W^\dagger. \tag{27}$$

## 3 Local Operator Entanglement

The entanglement of a time-evolving operator $O(t)$ is defined as the entanglement of the state $|O(t)\rangle$ corresponding to it under the state-to-operator mapping. Specifically, here we are interested in the entanglement of a connected real space region $A$ with respect to the rest of the system. Since the state corresponding to a time-evolving operator is pure, this quantity is conveniently measured by the Rényi entanglement entropies [61]

$$S_A^{(n)}(t) = \frac{1}{1-n} \log \text{tr}_A[\rho_A^n(t)], \tag{28}$$

where $\rho_A(t)$ is the density matrix at time $t$ reduced to the region $A$. Specifically, here we consider the evolution of the entanglement of the ultralocal operator $a_y$ and select half of the chain $A = [0, L/2)$. Moreover, here and in the following we will always take $a$ to be Hilbert-Schmidt orthogonal to the identity operator, i.e. traceless, to project out its trivial component.

With our choices of operator and subsystem the graphical representation for the reduced density matrix reads as

$$\rho_A(t, y; a) = \text{tr}_{\bar{A}}[|a_y(t)\rangle\langle a_y(t)|] = \qquad \qquad \qquad \qquad \qquad . \qquad (29)$$

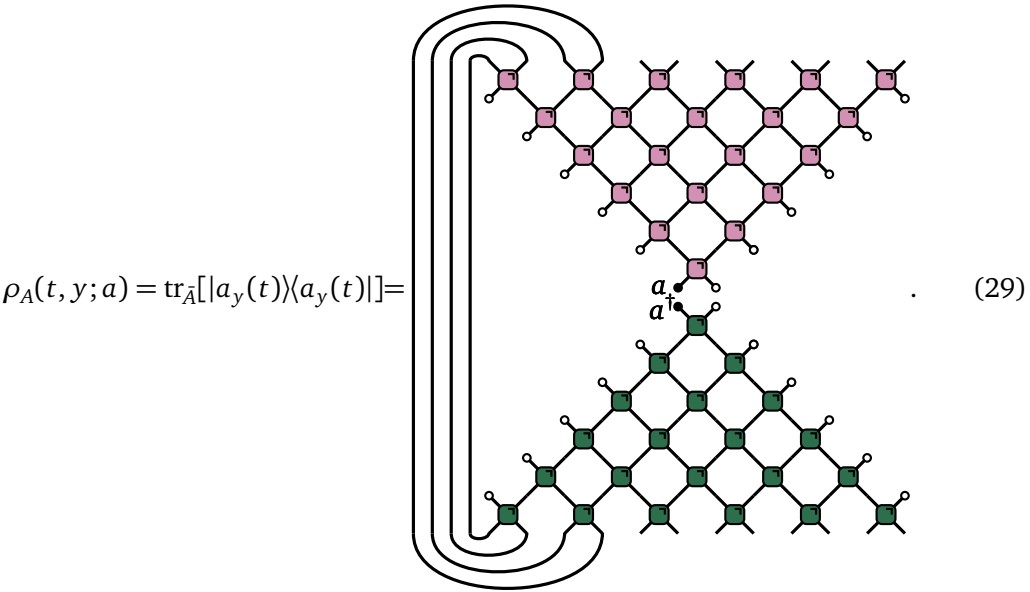

In the representation (29) we took $y < t \leq L$. We considered the right inequality, because we are interested in the thermodynamic limit and the results no longer depend on $L$ for $L \geq t$, while we take the left inequality, $t > y$, because in the opposite case the reduced density matrix is pure and hence the entanglement vanishes. This is due to the fact that in quantum circuits there is a strict lightcone for the propagation of information: nothing can propagate faster than a given maximal velocity (this is stricter than the Lieb-Robinson bound which allows for exponentially small corrections). In particular, in our units (see Eq. (20)) the maximal velocity is 1. Finally, we assumed $y$ to be an integer. The case of half-integer $y$ can be recovered by the reflection $R$ of the chain around the bond between 0 and 1/2. This results in

$$|a_y(t, U)\rangle\langle a_y(t, U)| \mapsto R|a_y(t, U)\rangle\langle a_y(t, U)|R^\dagger = |a_{1/2-y}(t, SUS^\dagger)\rangle\langle a_{1/2-y}(t, SUS^\dagger)| , \quad (30)$$

where $S$ is the "swap-gate"

$$S(a \otimes b)S^\dagger = b \otimes a , \qquad (31)$$

and we designate explicitly the dependence on the local gate. From now on we always take $y$ to be integer.

Using the representation (29) we see that the calculation of $\text{tr}_A[\rho_A^n(t, y; a)]$ is reduced to that of a partition function of a vertex model (generically with complex weights). For instance,

in the simplest nontrivial case $n = 2$ we have

$$\text{tr}_A[\rho_A^2(t, y; a)] = \quad\quad\quad\quad\quad\quad\quad\quad\quad\quad\quad\quad\quad\quad\quad\quad . \quad (32)$$

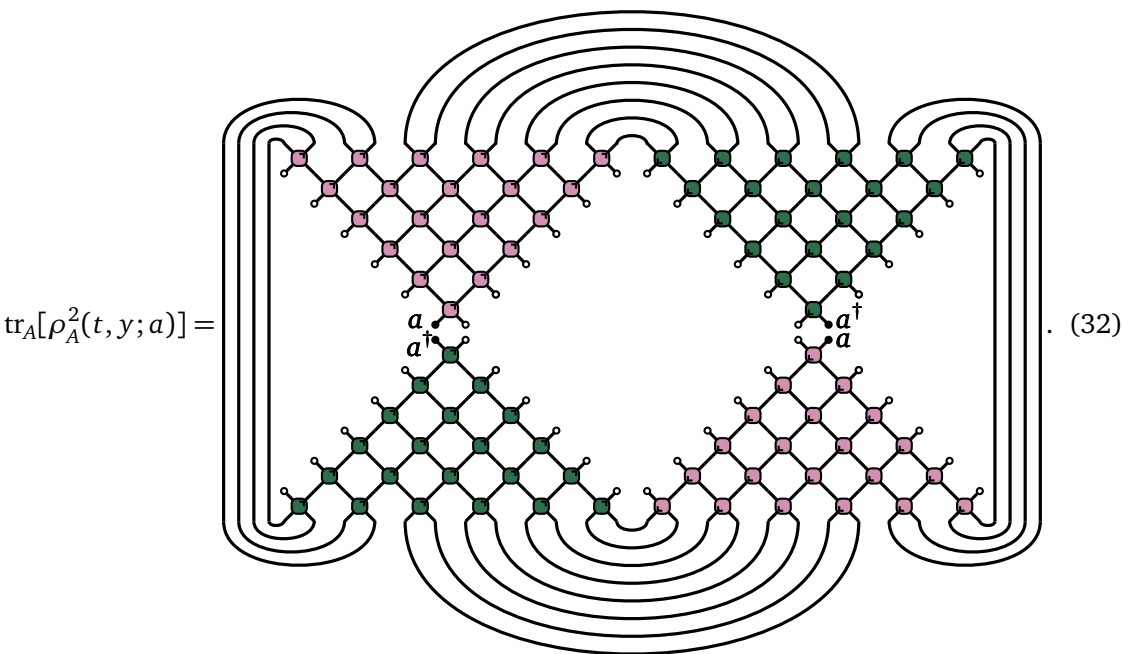

Using the graphical rules (26) this equation is reduced to

$$\text{tr}_A[\rho_A^2(t, y; a)] = \quad\quad\quad\quad\quad\quad\quad\quad\quad\quad\quad\quad\quad . \quad (33)$$

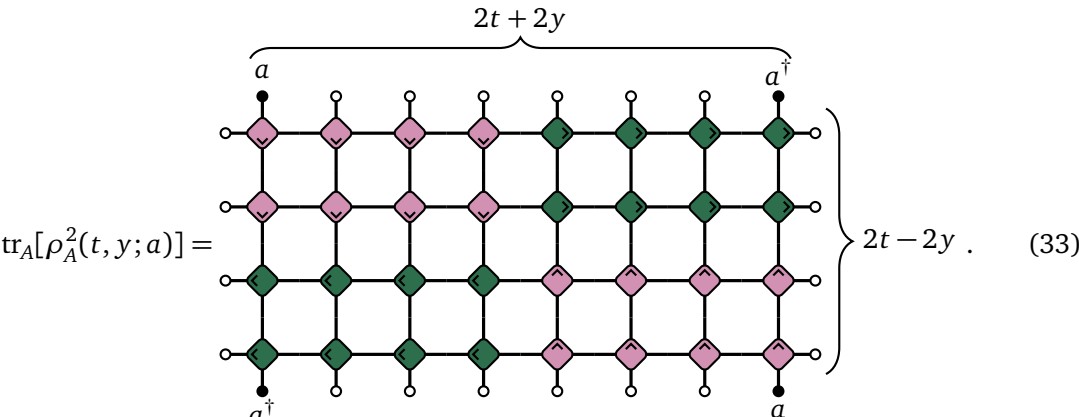

This expression can be rewritten as

$$\text{tr}_A[\rho_A^2(t, y; a)] = \|(\mathcal{H}_{x_+}[\mathbb{1}])^{x_-} |a^\dagger \underbrace{\circ \cdots \circ}_{2x_+ - 2} a\rangle \|^2 = \|(\mathcal{V}_{x_-}[\mathbb{1}])^{x_+ - 1} \mathcal{V}_{x_-}[a] |\underbrace{\circ \cdots \circ}_{2x_-}\rangle \|^2, \quad (34)$$

where we introduced the "light-cone coordinates"

$$x_+ \equiv t + y, \quad\quad\quad x_- \equiv t - y, \quad\quad\quad\quad (35)$$

and the row/column transfer matrices

$$\mathcal{H}_{x_+}[b] = \quad\quad\quad\quad\quad\quad\quad\quad\quad\quad\quad\quad\quad\quad b, \quad (36)$$

$$\mathcal{V}_{x_-}[b] = b$$ 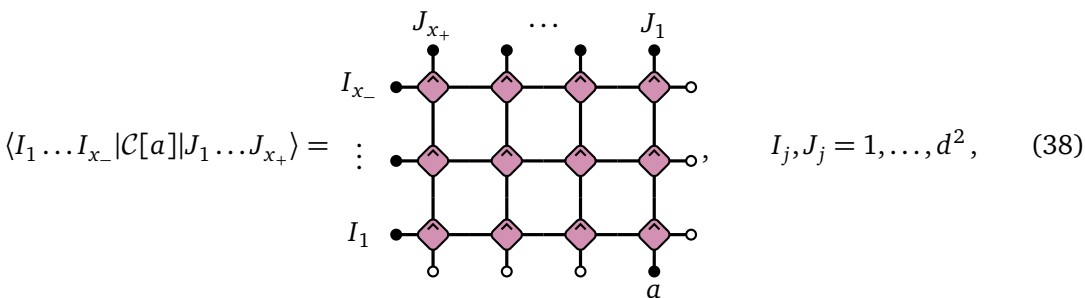 $$b^{\dagger}.$$ (37)

Note that $\mathcal{H}_{x_+}[b]$ is $d^{4x_+} \times d^{4x_+}$ while $\mathcal{V}_{x_-}[b]$ is $d^{4x_-} \times d^{4x_-}$ matrix.

Computing higher moments (i.e. higher Rényi orders) requires $n > 2$ replicas and involves partition functions on more complicated surfaces. In order to represent them compactly it is convenient to introduce the $d^{2x_-} \times d^{2x_+}$ "corner transfer matrix" [62, 63], defined by the following matrix elements

$$\langle I_1 \dots I_{x_-}|\mathcal{C}[a]\|J_1 \dots J_{x_+}\rangle = \qquad , \qquad I_j, J_j = 1, \dots, d^2, \qquad (38)$$

where $\{|I\rangle ; I = 1, 2, \dots, d^2\}$ is an orthonormal basis of $\mathcal{H}_1 \otimes \mathcal{H}_1$. Note that the corner transfer matrix is related to the row transfer matrices $\mathcal{H}_{x_+}[b]$ and $\mathcal{V}_{x_-}[b]$ as follows

$$\langle I_1 \dots I_{x_+}|\mathcal{C}[a]^{\dagger}\mathcal{C}[a]\ |J_1 \dots J_{x_+}\rangle = \langle I_1 \dots I_{x_+}, J_{x_+} \dots J_1|(\mathcal{H}_{x_+}[\mathbb{1}])^{x_-}|a^{\dagger} \circ \dots \circ a\rangle , \qquad (39)$$

$$\langle I_1 \dots I_{x_-}|\mathcal{C}[a]\ \mathcal{C}[a]^{\dagger}|J_1 \dots J_{x_-}\rangle = \langle I_1 \dots I_{x_-}, J_{x_-} \dots J_1|(\mathcal{V}_{x_-}[\mathbb{1}])^{x_+-1}\mathcal{V}_{x_-}[a]\| \circ \dots \circ\rangle . \qquad (40)$$

In terms of $\mathcal{C}[a]$ we can easily express the Rényi entropies as follows

$$S^{(n)}(y, t) = \frac{1}{1-n} \log \mathrm{tr}[(\mathcal{C}[a]^{\dagger}\mathcal{C}[a])^n] = \frac{1}{1-n} \log \mathrm{tr}[(\mathcal{C}[a]\mathcal{C}[a]^{\dagger})^n]. \qquad (41)$$

The problem of computing operator entanglement is then reduced to that of computing the moments of $\mathcal{C}[a]^{\dagger}\mathcal{C}[a]$ or of $\mathcal{C}[a]\mathcal{C}[a]^{\dagger}$.

Before concluding this section we note that under the gauge transformation (13) the traces of the reduced transfer matrix transform as follows

$$\mathrm{tr}_A[\rho_A(t, y; a)^n] \mapsto \begin{cases} \mathrm{tr}_A[\rho_A(t, y; u^{\dagger}au)^n] & y \in \mathbb{Z}_L - \frac{1}{2} \\ \mathrm{tr}_A[\rho_A(t, y; v^{\dagger}av)^n] & y \in \mathbb{Z}_L \end{cases} . \qquad (42)$$

This means that the gauge transformation only causes a rotation in the space of ultralocal operators.

# 4 Completely Chaotic Dual-Unitary Circuits

In this paper we consider *dual-unitary* circuits, i.e. local quantum circuits where the evolution remains unitary upon switching space and time directions. This means that the local two-qudit gate $U$ remains unitary if we consider left pair of wires as incoming states and right pair of wires as outgoing states. More formally, defining the "dual" (space) propagator $\tilde{U}$ by means of the relation

$$\langle i\,j|\,\tilde{U}\,|k\,l\rangle = \langle i\,k|\,U\,|j\,l\rangle , \qquad (43)$$

the circuit is *dual-unitary*, if both, $U$ and $\tilde{U}$ are unitary [56]. Dual unitarity can be expressed explicitly as

$$\sum_{p,q=1,2,\ldots,d} \langle \ell\, q|U^\dagger|k\, p\rangle \, \langle j\, p|U|i\, q\rangle = \delta_{\ell,i}\delta_{k,j}, \qquad \sum_{p,q=1,2,\ldots,d} \langle q\, \ell|U^\dagger|p\, k\rangle \, \langle p\, j|U|q\, i\rangle = \delta_{\ell,i}\delta_{k,j},$$
(44)

or diagrammatically as



$$(45a)$$

Considering the double gate (27), these relations lead to

$$(46a)$$

$$(46b)$$

We have shown in [56] that the dual-unitarity condition is not as stringent as one might think. For instance, in the case of qubits ($d = 2$) it only fixes two parameters of the sixteen specifying a generic matrix in $U \in U(4)$ and allows for a rich variety of dynamical behaviours [56]. Here, in particular, we focus on a specific class of dual-unitary circuits, which we term the *completely chaotic* class. To define it, we consider the transfer matrices $\mathcal{V}_x[\mathbb{1}]$ and $\mathcal{H}_x[\mathbb{1}]$. Since any such transfer matrix is a contracting operator, i.e.

$$\|\mathcal{T}\,|v\rangle\| \le \|\,|v\rangle\|, \qquad \mathcal{T} = \mathcal{V}_x[\mathbb{1}], \mathcal{H}_x[\mathbb{1}],$$
(47)

their eigenvalues are contained in the unit circle of the complex plane (see Appendix A for a proof of (47)). Using only the relations (46a) and (46b), we can find $x + 1$ independent simultaneous eigenvectors of $\mathcal{V}_x[\mathbb{1}]$ and $\mathcal{H}_x[\mathbb{1}]$ associated with eigenvalue one. They read as

$$\left\{|e_0\rangle = |\underbrace{\circ \ldots \circ}_{2x}\rangle,\ |e_1\rangle = |\underbrace{\circ \ldots \circ}_{x-1}\bar{r}_1\underbrace{\circ \ldots \circ}_{x-1}\rangle,\ \ldots,\ |e_{x-1}\rangle = |\circ\, \bar{r}_{x-1}\,\circ\rangle,\ |e_x\rangle = |\bar{r}_x\rangle\right\},$$
(48)

where we introduced the "rainbow" states $|r_l\rangle$ and their orthonormal counterparts $|\bar{r}_l\rangle$,

$$|r_l\rangle = \frac{1}{d^l}\sum_{I_1 I_2\ldots I_l=1}^{d^2}|I_1 I_2 \ldots I_l, I_l \ldots I_2 I_1\rangle = \overset{2l}{\overbrace{\text{...}}},$$
(49)

$$|\bar{r}_l\rangle = \frac{d}{\sqrt{d^2-1}}\left(|r_l\rangle - \frac{1}{d}|\circ\, r_{l-1}\,\circ\rangle\right),$$
(50)

satisfying $\langle \bar{r}_k | \bar{r}_l \rangle = \delta_{k,l}$. Note that the hermitian conjugates of these vectors are always left eigenvectors of $\mathcal{V}_x[\mathbb{1}]$ of $\mathcal{H}_x[\mathbb{1}]$ while (48) are right eigenvectors if the circuit is dual-unitary.

We are now in a position to introduce the following

**Definition 4.1.** *"Completely chaotic" dual-unitary circuits are the dual-unitary circuits such that (48) are the only eigenvectors of $\mathcal{V}_x[\mathbb{1}]$ and $\mathcal{H}_x[\mathbb{1}]$ associated to eigenvalues with unit magnitude.*

We stress that (48) are in general only a subset of the eigenvectors of $\mathcal{V}_x[\mathbb{1}]$ and $\mathcal{H}_x[\mathbb{1}]$ associated with eigenvalue one. For instance, integrable dual-unitary circuits (e.g. the one-parameter dual-unitary line of the integrable trotterised XXZ model [64], or the self-dual Kicked Ising model at the non-interacting point) have much more such eigenvectors (see Paper II for additional examples of such circuits). A thorough numerical analysis, however, proves that (i) *the completely chaotic class exists*; (ii) *it is the generic case*. In other words, generating a dual-unitary gate at random one would find with probability 1 that there are no eigenvectors of $\mathcal{V}_x[\mathbb{1}]$ and $\mathcal{H}_x[\mathbb{1}]$ with unit magnitude eigenvalues other than (48). The rest of the spectrum is gapped within a circle of radius strictly smaller than one.

Before moving on to the calculation of the local operator entanglement dynamics, it is interesting to investigate the relation between the definition 4.1 of completely chaotic circuits and the intuitive definition of chaos based on absence of local conservation laws. We will show that the class of completely chaotic dual-unitary circuits is in general more restrictive than that of chaotic ones. Namely, if a dual-unitary circuit has some non-trivial local conservation law $\mathcal{V}_x[\mathbb{1}]$ and $\mathcal{H}_x[\mathbb{1}]$ acquire some additional eigenvectors corresponding to the eigenvalue 1. In our discussion we will focus on circuits admitting conservation laws with local density which can be written either as

$$Q^- = \sum_{x \in \mathbb{Z}_L} q_x^-, \tag{51}$$

or as

$$Q^+ = \sum_{x \in \mathbb{Z}_L + \frac{1}{2}} q_x^+, \tag{52}$$

where the local densities $q_x^\pm$ act non-trivially (have support) on $r$ sites. More precisely, these densities act non-trivially on the intervals $[x, x+(r-1)/2] \cap \mathbb{Z}_L/2$ and $[x-(r-1)/2, x] \cap \mathbb{Z}_L/2$ respectively. Moreover, we choose the densities such that $\mathrm{tr}_x[q_x^\pm] = 0$ (here the trace is over the local Hilbert space at the $x$-th site). Note that this can be done without loss of generality: all charges can be written as combinations of $Q^+$ and $Q^-$.

Due to the two-site shift symmetry of the time evolution in the system we considered local conservation laws obtained by summing only on a sub-lattice (say even sites). In order for $Q^\pm$ to be conserved their local densities must satisfy continuity equations of the form

$$\mathbb{U}^\dagger q_x^- \mathbb{U} = q_x^- + J_{x-1}^- - J_x^-, \tag{53}$$

$$\mathbb{U}^\dagger q_{x-\frac{1}{2}}^+ \mathbb{U} = q_{x-\frac{1}{2}}^+ + J_{x-1}^+ - J_x^+, \qquad x \in \mathbb{Z}_L \tag{54}$$

for some "currents" $J_x^\pm$ supported on $r+1$ sites (for concreteness in writing (53) and (54) we assumed $r$ odd). As shown in Appendix B in dual-unitary circuits the relations (53) and (54) can be satisfied only if $J_x^- = q_x$ and $J_x^+ = -q_{x+\frac{1}{2}}$. This means that conserved-charge densities in dual-unitary circuits satisfy either

$$\mathbb{U}^\dagger q_x^- \mathbb{U} = q_{x-1}^-, \tag{55}$$

or

$$\mathbb{U}^\dagger q_{x-\frac{1}{2}}^+ \mathbb{U} = q_{x+\frac{1}{2}}^+, \qquad x \in \mathbb{Z}_L. \tag{56}$$

Let us show that these relations imply that $\mathcal{V}_x[\mathbb{1}]$ and $\mathcal{H}_x[\mathbb{1}]$ have additional eigenvectors corresponding to the eigenvalue 1. Focussing on the first and writing it in diagrammatic form for $r = 5$ we have

$$\text{(57)}$$

Tracing out the identities in the last three sites and repeatedly multiplying by the double gate $W$ we find

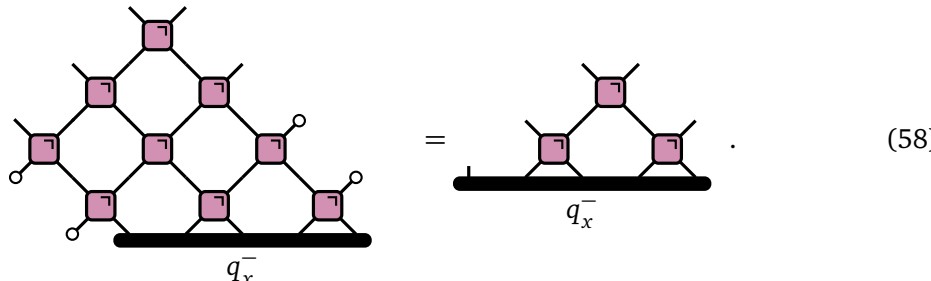

$$\text{(58)}$$

Finally, contracting the last two sites with $|\circ\circ\rangle$ we find

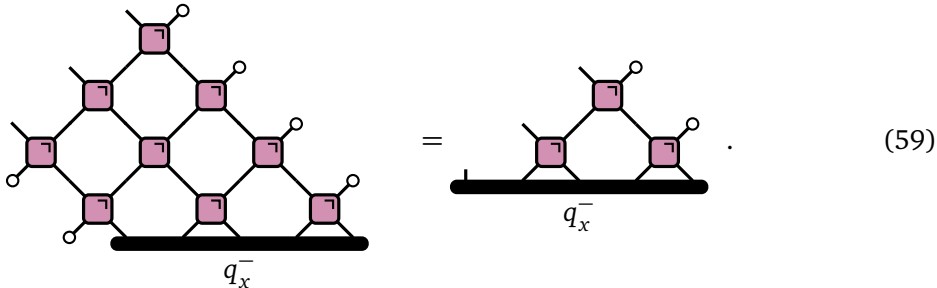

$$\text{(59)}$$

We then have that the vector

$$|v\rangle \equiv \qquad , \qquad \text{(60)}$$

is an eigenvector of $(V_3^{\circ\circ})^2$, where we introduced

$$V_x^{\circ\circ} = \qquad . \qquad \text{(61)}$$

Note that $|v\rangle$ cannot be zero. Indeed, if this were to be the case also the l.h.s. of (58) would vanish leading to an absurd: the r.h.s. of that equation features the non-zero operator $q_x^-$ conjugated by unitary matrices. To conclude or argument we note that

$$|w\rangle = |v\rangle + V_x^{\circ\circ}|v\rangle \qquad \text{(62)}$$

is an eigenvector of $V_x^{\circ\circ}$ corresponding to the eigenvalue 1. Then, we can construct many additional eigenvectors of $\mathcal{V}_{x \geq r}[\mathbb{1}]$ corresponding to eigenvalue 1, for example

$$\left\{ |w \underbrace{\circ \ldots \circ}_{2x-r}\rangle, |\circ w \underbrace{\circ \ldots \circ}_{2x-r-1}\rangle, \ldots, |\underbrace{\circ \ldots \circ}_{x-r-1} w \underbrace{\circ \ldots \circ}_{x+1}\rangle, |\underbrace{\circ \ldots \circ}_{x-r} w \underbrace{\circ \ldots \circ}_{x}\rangle \right\}. \tag{63}$$

Finally we note that, if (55) holds, the charges

$$Q^-_{k;s_1,\ldots,s_k} = \sum_{x \in \mathbb{Z}_L} q^-_x q^-_{x+s_1} \cdots q^-_{x+s_k}, \qquad k = 1, \ldots, \; s_j \in \mathbb{Z}_L, \tag{64}$$

are also conserved. Considering the densities of such charges and proceeding as before we can construct exponentially many (in $x$) eigenvectors of $\mathcal{V}_{x \geq r}[\mathbb{1}]$ with eigenvalue 1. An analogous reasoning considering a conserved density $q^+_x$ would instead produce additional eigenvectors of $\mathcal{H}_{x \geq r}[\mathbb{1}]$ corresponding to eigenvalue 1.

## 5   Dynamics of Local Operator Entanglement

It is generically very difficult to say much about the dynamics of local operator entanglement in interacting systems (in fact, this quantity is generically out of reach even in the presence of integrability). Here we show that in completely chaotic dual-unitary circuits one can make some quantitative progress.

In the first part, we prove that in the two limits $x_\pm \to \infty$ the local operator entanglement can be determined *exactly*. These two limits correspond to varying the initial position of the operator in order to measure the entanglement generated at the edges of the light-cone ($x_- \to \infty$ gives the entanglement generated by the right edge and $x_+ \to \infty$ that generated by the left: see Fig. 1 for a pictorial representation). Note that, the operator "breaks" the left-right symmetry of the problem and one should not expect the results of the two limits to coincide. Indeed, we find that they are physically *very different*. In particular, while the entanglement generated by the right edge has *flat spectrum* and grows at the *maximal speed*, the one generated by the left edge is much richer. First it has a *non-trivial spectrum* and second, while the von Neumann entropy always grows at the maximal speed, higher Rényi entropies show a *phase transition* in the speed of the entanglement growth when varying the parameters of the gate. Specifically the growth depends on the largest eigenvalue $\lambda$ governing the decay of the dynamical correlations (cf. [56]).

In the second part of the section we show that the "local operator $n$-purities"

$$e^{(1-n)S^{(n)}(y,t)} = e^{(1-n)S^{(n)}((x_+-x_-)/2,(x_++x_-)/2)} = \mathrm{tr}_A[\rho^n_A(t)], \tag{65}$$

for *any* $x_+$ and $x_-$ are well described (even at short times) by summing the two limits $x_\pm \to \infty$, namely

$$e^{(1-n)S^{(n)}(y,t)} \approx \lim_{x_- \to \infty} e^{(1-n)S^{(n)}(y,t)} + \lim_{x_+ \to \infty} e^{(1-n)S^{(n)}(y,t)} \qquad n > 1. \tag{66}$$

This indicates that in completely chaotic dual unitary circuits the bulk of the light-cone region rapidly becomes highly entangled (and hence it does not contribute to the purities) and the leading contributions to the purities arises from the edges. Interestingly, if the dynamical correlations of a circuit decay fast enough we observe the local operator entanglement growing at the maximal speed ($\log d^2$); otherwise the growth is slower and depends on $\lambda$. In the latter case the entanglement spectrum is non-trivial.

Finally we extend the above results to a class of chaotic but not completely chaotic dual unitary circuits including the self dual Kicked Ising model. In particular, we compute exactly

the operator entanglement in the two limits $x_\pm \to \infty$ and, by comparing with numerical simulations, we show that the property (66) continues to hold far enough from integrable points.

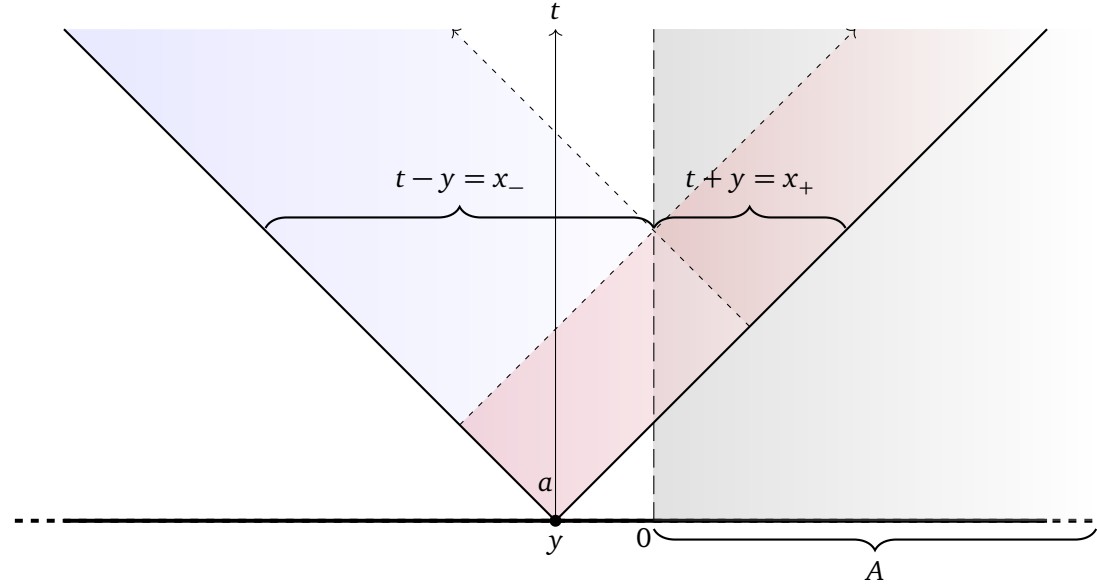

Figure 1: Pictorial illustration of the operatorial evolution, depicting the two limits (72) and (75). The two limits are taken along the dashed arrows. The first limit $x_- \to \infty$ has $x_+$ constant, which means that as time increases we move the operator to the left getting contribution from the red region. The second limit $x_+ \to \infty$ has $x_-$ constant, meaning that, as time increase we move the operator to the right and get contribution from the blue region.

## 5.1 The Two Limits

Let us start by considering the special limits described above, where one focusses on the entanglement generated by the edge of the light-cone produced by the spreading operator $a$.

### 5.1.1 The Limit $x_- \to \infty$

Let us first consider the entanglement generated by the right light-cone edge, namely we consider the limit $x_- = t - y \to \infty$ while keeping $x_+ = t + y$ fixed. In this limit it is convenient to use the representation (39). We start by nothing that, since the operator $a$ is traceless (cf. Sec. 3), the only eigenvector of $\mathcal{H}_x[\mathbb{1}]$ with non-zero overlap with the "initial state" $|a^\dagger, \circ, ..., \circ, a\rangle$ is $|e_{x_+}\rangle = |\bar{r}_{x_+}\rangle$. This is because the scalar product $\langle e_{x_+}|a^\dagger \circ \cdots \circ a\rangle$ is the only one among $\{\langle e_i|a^\dagger \circ \cdots \circ a\rangle\}_{i=1,...,x_+}$ which does not produce the trace of $a$. In particular

$$\langle e_{x_+}|a^\dagger \underbrace{\circ \cdots \circ}_{2x_+ - 2} a\rangle = \frac{d}{\sqrt{d^2-1}} \langle r_{x_+}|a^\dagger \underbrace{\circ \cdots \circ}_{2x_+ - 2} a\rangle = \frac{d^{1-x_+}\text{tr}[a^\dagger a]}{\sqrt{d^2-1}} = \frac{d^{1-x_+}}{\sqrt{d^2-1}}. \qquad (67)$$

Where we used that $a$ is Hilbert-Schimdt normalised. Plugging in the definition of $\mathcal{C}[a]^\dagger \mathcal{C}[a]$ we then have

$$\lim_{x_- \to \infty} \mathcal{C}[a]^\dagger \mathcal{C}[a] = \langle e_{x_+}|a^\dagger \circ \cdots \circ a\rangle M_{x_+} = \frac{d^{1-x_+}}{\sqrt{d^2-1}} M_{x_+}, \qquad (68)$$

where we introduced the $d^{2x_+} \times d^{2x_+}$ matrix $M_k$ ($k \in \{0, 1, \ldots, x_+\}$) defined as

$$\langle I_1 \ldots I_{x_+} | M_k | J_1 \ldots J_{x_+} \rangle = \langle I_1 \ldots I_{x_+}, J_{x_+} \ldots J_1 | e_k \rangle. \tag{69}$$

$M_k$ can be expressed in matrix form as

$$M_k = \left( \frac{d^{1-k}}{\sqrt{d^2-1}} \right)^{1-\delta_{k,0}} P_{1,\circ} \ldots P_{x_+-k,\circ}(\mathbb{1}^{\otimes x_+} - P_{x_+-k+1,\circ}(1-\delta_{k,0})), \tag{70}$$

where $\mathbb{1}$ denotes the identity element in $\text{End}(\mathcal{H}_1 \otimes \mathcal{H}_1)$ and

$$P_{k,\circ} = \mathbb{1}^{\otimes(k-1)} \otimes |\circ\rangle\langle\circ| \otimes \mathbb{1}^{\otimes(x_+-k)}, \tag{71}$$

is a projector to the state $\circ$ on site $k$. Plugging (68) into (41) we find that the end result for a Rényi entropy of generic order $n$ reads

$$\lim_{x_- \to \infty} S^{(n)}(y, t) = \lim_{x_- \to \infty} \frac{1}{1-n} \log \text{tr}_A[\rho_A^n(t, y; a)] = x_+ \log d^2 - \log \left( \frac{d^2}{d^2-1} \right), \tag{72}$$

where, to take the trace, we used

$$\text{tr} (M_k)^n = \left( \frac{d^{1-k}}{\sqrt{d^2-1}} \right)^{n-2} \quad k > 0, \qquad \text{tr} (M_0)^n = 1. \tag{73}$$

Eq. (72) gives linear growth of the operator entanglement entropy with the maximal slope, and holds in the absence of "non-generic" eigenvectors of eigenvalue 1.

### 5.1.2 The Limit $x_+ \to \infty$

Let us now consider the limit $x_+ = t + y \to \infty$ while keeping $x_- = t - y$ fixed. This limit can be evaluated using (40) but we immediately see that it is more complicated than the previous one: we need to deal with the operator-dependent transfer matrix $\mathcal{V}_{x_-}[a]$ and all of $x_- + 1$ eigenvectors. The calculation yields

$$\lim_{x_+ \to \infty} \mathcal{C}[a] \, \mathcal{C}[a]^\dagger = \sum_{k=0}^{x_-} M_k \, \langle e_k | \mathcal{V}_{x_-}[a] | \underbrace{\circ \cdots \circ}_{2x_-} \rangle. \tag{74}$$

Therefore we obtain

$$\lim_{x_+ \to \infty} S^{(n)}(y, t) = \frac{1}{1-n} \log \left[ \sum_{k=0}^{x_-} |\langle e_k | \mathcal{V}_{x_-}[a] | \underbrace{\circ \cdots \circ}_{2x_-} \rangle|^n \left( \frac{d^{2-2k} - \delta_{k,0}}{d^2-1} \right)^{n/2-1} \right]. \tag{75}$$

Once again, this result holds in the absence of additional eigenvectors of $\mathcal{V}_{x_-}[\mathbb{1}]$ with unit magnitude, i.e. for completely chaotic dual-unitary circuits.

The missing information in (75) is the value of $\langle e_k | \mathcal{V}_{x_-}[a] | \circ \cdots \circ \rangle$, which can be expressed in terms of

$$\langle \circ \ldots \circ r_l \circ \ldots \circ | \mathcal{V}_{x_-}[a] | \underbrace{\circ \cdots \circ}_{2x_-} \rangle =$$

$$\tag{76}$$

This expression can be evaluated by writing the elements of the sum using the single qudit map introduced in [56] for calculating the dynamical correlation functions. The central part of (76), which results from the contraction with the rainbow state $|r_l\rangle$, simplifies due to the unitarity of the gate, and produces a factor $d^{-l}$. The rest of the expression can be written in terms of the maps

$$\langle b|\mathcal{M}_{+,U}|a\rangle = a \!\!\bullet\!\!\Diamond\!\!\bullet\! b^\dagger \qquad \text{and} \qquad \langle b|\mathcal{M}_{-,U^\dagger}|a\rangle = a \!\!\bullet\!\!\Diamond\!\!\bullet\! b^\dagger, \qquad (77)$$

as

$$\langle e_l|\mathcal{V}_{x_-}[a]|\underbrace{\circ\cdots\circ}_{2x_-}\rangle = \frac{d^{1-l}}{\sqrt{d^2-1}}\left(\langle a|\mathcal{M}_{-,U^\dagger}^{x_--l}\mathcal{M}_{+,U}^{x_--l}|a\rangle - \langle a|\mathcal{M}_{-,U^\dagger}^{x_--l+1}\mathcal{M}_{+,U}^{x_--l+1}|a\rangle\right) \quad l>0, \quad (78)$$

$$\langle e_0|\mathcal{V}_{x_-}[a]|\underbrace{\circ\cdots\circ}_{2x_-}\rangle = \langle a|\mathcal{M}_{-,U^\dagger}^{x_-}\mathcal{M}_{+,U}^{x_-}|a\rangle. \qquad (79)$$

The maps can be expressed using a $d^2 \times d^2$ matrix and the expressions are then easily evaluated numerically. Moreover, the maps $\mathcal{M}_{-,U^\dagger}$ and $\mathcal{M}_{+,U}$ have the same eigenvalues and their respective eigenvectors $|e_{-,U^\dagger}\rangle$ and $|e_{+,U}\rangle$ are connected via the relation $|e_{+,U}\rangle = v_+^* \otimes v_+ |e_{-,U^\dagger}\rangle$ ($v_+$ is part of the parametrisation of $U$ cf. Appendix C).

The leading asymptotic behaviour is governed by the leading eigenvalue[2] $\lambda$ ($|\lambda| \leq 1$) of the map $\mathcal{M}_{+,U}$ and can be determined analytically by posing

$$\langle a|\mathcal{M}_{-,U^\dagger}^l\mathcal{M}_{+,U}^l|a\rangle = |\lambda|^{2l}c_l, \qquad (80)$$

in (78) and (79). Here $c_l$ is bounded in $l$, i.e.

$$\limsup_{l\to\infty} c_l < \infty. \qquad (81)$$

Plugging in (75) we find the following asymptotic result

$$\Delta S^{(n)}|_{\mathrm{asy},x_+} = \lim_{x_-\to\infty}\lim_{x_+\to\infty} S^{(n)}(y,t)/x_- = \begin{cases} \log d^2, & |\lambda| < d^{\frac{1-n}{n}} \\ \log|\lambda|^{\frac{2n}{1-n}}, & d^{\frac{1-n}{n}} \leq |\lambda| < 1 \end{cases}. \qquad (82)$$

The result is intriguing, we see a transition between maximal and a sub-maximal growth, governed by the slowest decay of the two point dynamical correlation functions. Moreover, we see that the entanglement spectrum is not flat in this limit, but the result encodes a non-trivial $n$-dependence, see Fig. 2. This is very different from the limit $x_- \to \infty$, where all entropies experience maximal growth. Furthermore, there is another interesting observation to make. Performing an analytical continuation of the result in $n$ and taking the limit $n \to 1^+$ we find that the the growth of von-Neumann entropy ($n \to 1^+$) is *always maximal*.

## 5.2 The Conjecture

Let us now consider the local operator entanglement for *generic* $x_-$ and $x_+$. To describe its leading in time behaviour we propose the following conjecture

**Conjecture 5.1.** *For chaotic dual-unitary local circuits, at long times the operator entanglement entropies for $n > 1$ are well described by the sum of the two limits (72) and (75). Namely*

$$S^{(n)}(y,t) \approx \frac{1}{1-n}\log\left[\lim_{x_-\to\infty} e^{(1-n)S^{(n)}(y,t)} + \lim_{x_+\to\infty} e^{(1-n)S^{(n)}(y,t)}\right], \qquad n > 1. \qquad (83)$$

---

[2]Excluding the trivial eigenvalue 1.

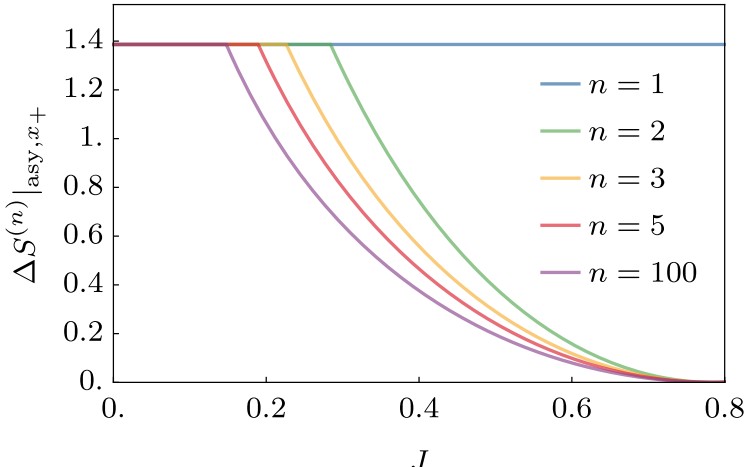

Figure 2: The asymptotic slope $\Delta S^{(n)}|_{\mathrm{asy},x_+}$ (82) as a function of the gate parameter $J$ (see Appendix C for details on the parametrisation) for different values of $n$ (different colors). The slope is $n$-independent in the maximal-growth region but both the size of this region and the slope away from it depend on $n$.

$$\mathrm{tr}\left(\!\!\begin{array}{c}\vcenter{\hbox{\includegraphics{fig}}}\end{array}\!\!\right)^{\!n}_{\underbrace{\phantom{xx}}_{x_-}\underbrace{\phantom{xx}}_{x_+}} = \mathrm{tr}\left(\!\!\begin{array}{c}\vcenter{\hbox{\includegraphics{fig}}}\end{array}\!\!\right)^{\!n}_{\underbrace{\phantom{xx}}_{x_-}} + \mathrm{tr}\left(\!\!\begin{array}{c}\vcenter{\hbox{\includegraphics{fig}}}\end{array}\!\!\right)^{\!n}_{\underbrace{\phantom{xx}}_{x_+}}$$

Figure 3: Pictorial illustration of the Conjecture 5.1. The entanglement of the spreading operator at time $t$ is written as a sum of two contributions where the left and right boundary are respectively sent to infinity.

As pictorially represented in Fig. 3, the conjecture consists in replacing the trace of the $n$-th power of the reduced density matrix of the "vectorised" spreading operator $a_y(t)$ with the sum of two terms. These terms are the trace of $n$-th powers of density matrices corresponding to operators obtained from $a_y(t)$ by sending to infinity respectively its left $(-x_-)$ or right $(x_+)$ edges. Note that the conjecture cannot hold for the von-Neumann entropy, as the limit $n \to 1^+$ of the r.h.s. of (83) is singular (the argument of the logarithm goes to 2).

Conjecture 5.1 yields the following form for the entanglement entropies

$$S^{(n)}(y,t) = t\Delta S^{(n)}(y,t) + \mu_n(y,t), \tag{84}$$

where the "slope" $\Delta S^{(n)}(y,t)$ and the "offset" $\mu_n(y,t)$ are bounded in $t$. We evaluated Eq. (84) using Eq. (72) and Eq. (75) and compared it to the results of exact short-time numerical simulations — obtained by direct diagonalisation of the corner transfer matrix (see Appendix E for details). The comparison, for $n = 2$ and $y = 0$, is reported in Fig 4. The figure presents results for both the slope $\Delta S^{(2)}(0,t)$ and the constant shift $\mu_2(0,t)$, which is very sensitive to small errors in the slope. The agreement observed is remarkable, even for the short times accessible by the numerics. A similar level of agreement is observed also for $n \geq 2$.

The asymptotic value of the slopes in the limit $t \to \infty$ with fixed "ray" $\zeta = y/t$ are given

by

$$\lim_{\substack{t \to \infty \\ y/t = \zeta}} \Delta S^{(n)}(y,t)/t = \begin{cases} \max\left[(1-\zeta)\log d^2, (1+\zeta)\log d^2\right], & |\lambda| < d^{\frac{1-n}{n}} \\ \max\left[(1-\zeta)\log|\lambda|^{\frac{2n}{1-n}}, (1+\zeta)\log d^2\right], & d^{\frac{1-n}{n}} \leq |\lambda| < 1 \end{cases}. \tag{85}$$

Equation (85) predicts a "phase transition" between a region with slopes that are symmetric in $\zeta$ (as it happens for random unitary circuits, see Ref. [37]) and a region where instead they show an interesting asymmetry in $\zeta$, and, moreover, they become $n$-dependent. In particular, we see that for $\zeta = 0$ the slopes coincide with those given in (82) and the slope in the symmetric region is the maximal one ($\log d^2$ is the maximal entanglement growth attainable in a circuit with $d$-dimensional local Hilbert space). For comparison, we computed numerically the dynamics of the local operator Rényi-2 entropy in Haar-random non-dual-unitary local qubit circuits for $\zeta = 0$. We considered two cases: (i) we chose the same (constant) gate for all the space-time points (clean case); (ii) we took different i.i.d. $U_{x,t}$ for each space-time point in the circuit (noisy case). In both cases we obtained roughly half of the maximal slope of the entanglement entropy growth, see Fig. 5. This is in accordance with the predictions of Ref. [37] and proves that, in some parameter ranges, dual-unitary circuits are "more chaotic" than the average.

The idea behind Conjecture 5.1 is most easily explained considering the purity $e^{-S^{(2)}(y,t)}$. Looking at the representation (33) we see that this quantity can be written as the partition function of a statistical mechanical model (with complex weights) on a rectangle of dimensions $2x_+$ and $2x_-$. The conjecture corresponds to restricting this partition function to the sum over configurations spanning eigenvectors of eigenvalue 1 of both row and column transfer matrices. The same idea applies to $n$-purities with $n > 2$. Physically, this corresponds to assume that the bulk of the light-cone is highly scrambled (i.e. it gives a very small contribution to the purity), while the regions at a finite distance from the light-cone edges present the minimal scrambling (i.e. give the leading contribution to the purity). This is justified by noting that close to the light-cone edge the operator retains the maximal amount of information on the initial condition. We expect this picture to hold true for more general, non-dual unitary, chaotic systems if one replaces the light cone spreading at the maximal speed (1 in our units) with an effective one spreading at the "butterfly velocity" $v_B$ [6] of the system. Indeed, $v_B$ is by definition the velocity at which the scrambled region spreads in time.

## 5.3 Self-Dual Kicked Ising Model ($d = 2$)

Conjecture 5.1 is assumed to describe the asymptotic dynamics of the local operator entanglement in any chaotic dual-unitary circuit. In order for it to have any predictive power, however, one must be able to compute the limits $x_\pm \to \infty$. While in the previous subsections we showed that this can be done for the completely chaotic subclass, here we show that the limits can be computed exactly also in the paradigmatic example of dual-unitary circuits in $d = 2$: the self-dual kicked Ising model [57, 58]. This model is not completely chaotic according to Definition 4.1 because it possesses additional structure. Specifically, its local gate fulfils

$$U^\dagger(\alpha \otimes \mathbb{1})U = w \otimes \alpha, \qquad U^\dagger(\mathbb{1} \otimes \alpha)U = \alpha \otimes w, \tag{86}$$

where $w, \alpha$ are some hermitian and traceless matrices in SU(2)[3]. This condition leads to $x$ additional eigenvectors of the transfer matrices $\mathcal{V}_x[\mathbb{1}]$ and $\mathcal{H}_x[\mathbb{1}]$ with eigenvalue one. In fact, as shown in Appendix D, all reflection symmetric dual-unitary circuits fulfilling (86) are gauge equivalent to the self-dual kicked Ising.

---

[3]$w, \alpha$ can actually be any $2 \times 2$ complex matrices, from which we can derive analogous relations with traceless hermitian matrices (cf. Lemma A.1. of Paper II).

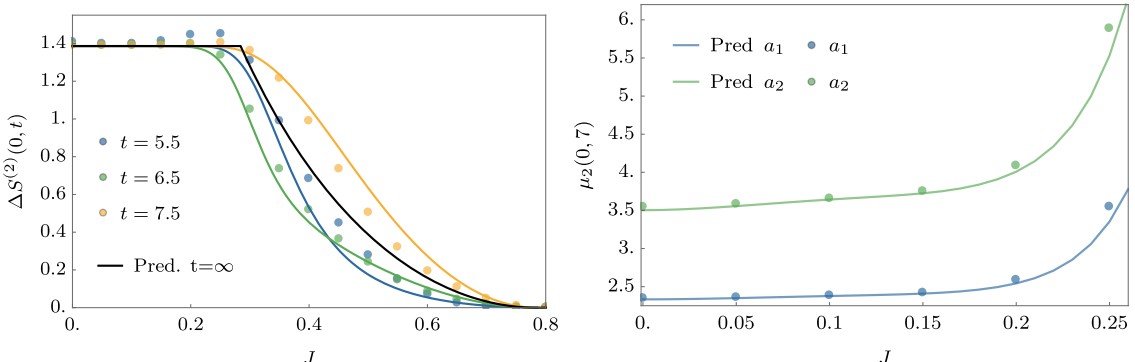

Figure 4: The slope $\Delta S^{(2)}(y, t-1/2) = S^{(2)}(y, t) - S^{(2)}(y, t-1)$ and the constant offset factor $\mu_2(y, t)$ (much more sensitive) versus the parameter $J$ for a dual unitary gate with $r = 0.5$, $\phi = 0.7$, $\theta = 0$ (see Appendix C for the definition of the gate). We show the results for operators $a_1 = \sigma_3$ (left panel) and $a_2 = \alpha_1 \sigma_1 + \alpha_2 \sigma_2 + \alpha_3 \sigma_3$ a fixed random operator with $\alpha_1 = 0.3289$, $\alpha_2 = 0.0696$, $\alpha_3 = 0.6221$. The points correspond to exact numerical results, and the lines are the predictions using the conjecture (83). The operator is initialised at $y = 0$, and we set $t = 7$ for the right panel.

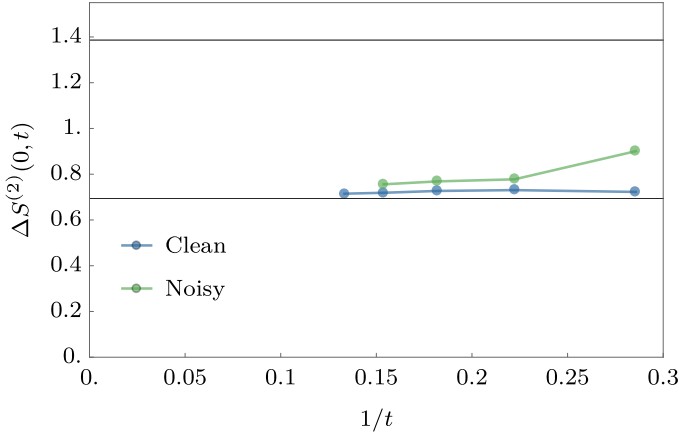

Figure 5: The slope of Rényi $n = 2$ entanglement entropy for the operator $\sigma_3$ evolving according to (non-dual-unitary) $U(4)$ Haar-random gates. In the clean case we average over 10 realisations, in the noisy case over 20 (100 for $t \leq 6$). The results suggest that the slope is close to $\log 2$, which is half of the maximal slope. Note that this agrees well with large $d$ result from [37], where we get the slope $2\frac{6}{5}(\sqrt{2}-1)\log 2 \approx 0.9941 \log 2$, if we use the parameters $s_{spread}, v_b$ for $d = 2$ (cf. Ref. [37] for a definition of these parameters).

We can use the gauge transformation (13) to set $\alpha = \sigma_3$, which holds in the standard formulation of self-dual kicked Ising model. The additional eigenvectors with eigenvalue one are then given by

$$\left\{ |e'_{x+1}\rangle = |\circ \ldots \circ 3\, 3\, \underbrace{\circ \ldots \circ}_{x-1}\rangle,\ |e'_{x+2}\rangle = |\circ \ldots \circ 3\, r_1\, 3\, \underbrace{\circ \ldots \circ}_{x-2}\rangle, \ldots, |e'_{2x}\rangle = |3\, r_{x-1}\, 3\rangle \right\}, \tag{87}$$

where 3 stands for the operator $\sigma_3$. To construct an orthonormal basis, we consider the fol-

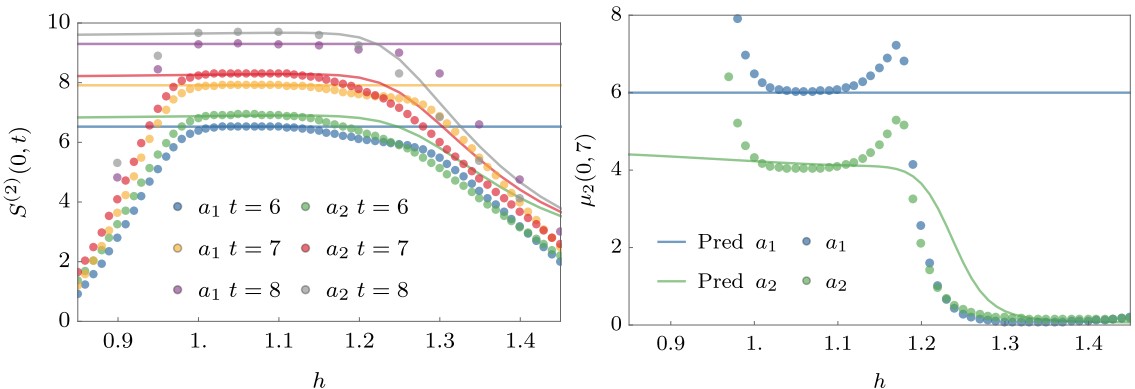

Figure 6: Prediction and numerical data for self-dual kicked Ising model. The points correspond to exact results, and the lines are the predictions using the conjecture (83). In the left panel we show the Rényi $n = 2$ entropy at times $6, 7, 8$ versus the magnetic field parameter $h$. The deviations from the prediction observed away from the central region are due to the vicinity of solvable points $h = k\frac{\pi}{4}$, $k \in \mathbb{Z}$, where the model is a Clifford circuit. There the prediction fails, because the transfer matrix has additional eigenvectors of eigenvalue 1. But as time increases, the region where the prediction holds grows. In the right panel we show the more sensitive constant offset factor $\mu_2$ versus the parameter $h$ at $t = 7$. The operators $a_1, a_2$ are the same as those used in Fig. 4 and they are initialised in $y = 0$.

lowing linear combination

$$|e_{x+i}\rangle = \sqrt{\frac{3}{2}} |e'_{x+i}\rangle - \frac{1}{\sqrt{2}} |e_i\rangle \qquad i \in \{1, \dots, x\}. \tag{88}$$

Having the eigenvectors, we evaluate the limits:

$$\lim_{x_- \to \infty} S^{(n)}(y, t) = (x_+ - 1) \log 4 - \log\left(\frac{1}{2} \left||\alpha_x|^2 + |\alpha_y|^2\right|^2 + |\alpha_z|^4\right), \tag{89}$$

$$\lim_{x_+ \to \infty} S^{(n)}(y, t) = -\log \sum_{j=0}^{2x_-} |\langle e_j| \mathcal{V}_{x_+}[a] \underbrace{|\circ \cdots \circ\rangle}_{2x_-}|^2 = -\log \sum_{j=0}^{x_-} \frac{3 - \delta_{j,0}}{2} |\langle e_j| \mathcal{V}_{x_+}[a] \underbrace{|\circ \cdots \circ\rangle}_{2x_-}|^2, \tag{90}$$

where we parametrised the initial local operator as

$$a = \alpha_1 \sigma_1 + \alpha_2 \sigma_2 + \alpha_3 \sigma_3, \qquad |a_1|^2 + |a_2|^2 + |a_3|^2 = \frac{1}{2}. \tag{91}$$

The last equality in Eq. (90) follows from $\langle e'_j| \mathcal{V}_{x_+}[a] |\circ \cdots \circ\rangle = 0$. Therefore, the additional eigenvectors change only the constant prefactors. Equations (89) and (90) show that, in the long time limit, the offset constant $\mu_2$ is different with respect to the result in completely chaotic dual-unitary circuits (cf. Eqs. (72) and (75)), but the slope is the same.

With the limits $x_\pm \to \infty$ at hand we are now in a position to compare the prediction of Conjecture 5.1 with (short-time) numerics. A comparison is shown in Fig. 6. The figure shows that, far enough from some special points in parameter space (see the caption), there is good agreement even for numerically accessible times ($t \leq 8$).

# 6 Conclusions

In this paper we studied the local operator entanglement growth in dual-unitary circuits. We identified a completely chaotic class for which the local operator entanglement always grows linearly in time. For this class we provided a quantitative description of the local-operator-entanglement dynamics based on a simple conjecture, which is strongly supported by numerical results. We postulated that, at late enough times, the local operator purities (traces of powers of the reduced density matrix) can be determined by considering separately the entanglement produced by the two edges of the spreading operator and then summing them together. In other words, we wrote the exponentials of the operator entanglement entropies as sums of two contributions, respectively obtained by sending the right edge of the spreading operator to infinity and the left edge to minus infinity. Our conjecture, together with the dual-unitarity property, allows to evaluate *analytically* the local operator entanglement of generic operators initially localised on a single site. These results have been extended to the self-dual kicked Ising model (which does not fall into the completely chaotic class). We argued that a modified form of our conjecture should hold in generic chaotic systems, i.e. for non-dual-unitary circuits, however, without dual-unitarity it does not directly yield analytical predictions.

Interestingly, our conjecture predicts that the slope of the local operator entanglement displays an abrupt transition when varying the parameters of the circuits. Moreover, the point in which the transition occurs depends on the Rényi index. On one side of the transition the slope of growth is the maximal allowed by the geometry of the circuit ($\log d^2$), which is approximately twice as large as that observed in Haar-random circuits [37]. This indicates that a subset of our chaotic dual-unitary circuits can be regarded as minimal solvable models for the maximally chaotic dynamics. On the contrary, on the other side of the transition the slope is not maximal, depends on the Rényi index, and approaches 0 when the dual unitary gate approaches the SWAP gate.

Our work raises a number of questions that can guide future research. First, our conjecture seems to describe the numerics even at small times, suggesting that it holds up to very small corrections. It would be interesting to investigate this aspect further and, possibly, rigorously prove the conjecture. Second, the class of systems that we introduced here (see also [56]) can be used to study exactly many aspects of non-equilibrium dynamics in chaotic systems, from relaxation of local observables to the behaviour of out-of-time-ordered correlations.

# Acknowledgements

We thank Lorenzo Piroli, Marko Medenjak, Vincenzo Alba, Jérôme Dubail, Andrea De Luca, and Tibor Rakovszky for useful discussions. BB thanks LPTMS Orsay for hospitality during the completion of this work.

**Funding Information** This work has been supported by the European Research Council under the Advanced Grant No. 694544 – OMNES, and by the Slovenian Research Agency (ARRS) under the Programme P1-0402.

# A Row and Column Transfer Matrices are Contracting

In this appendix we show that the eigenvalues of row, $\mathcal{H}_x[\mathbb{1}]$, and column, $\mathcal{V}_x[\mathbb{1}]$, transfer matrices (cf. (36) and (37) respectively) have absolute value bounded by 1. Let us show for

one of them, say $\mathcal{V}_x[\mathbb{1}]$, as showing it for the other one is completely analogous. Considering the expectation value of $\mathcal{V}_x[\mathbb{1}]$ on a generic state $|\psi\rangle \in (\mathbb{C}^d)^{\otimes 2x}$ we have

$$|\langle\psi|\mathcal{V}_x[\mathbb{1}]|\psi\rangle| = |\langle\circ\psi|\mathfrak{U}|\psi\circ\rangle|, \qquad (92)$$

where $\mathfrak{U} = W_{0,1/2}\cdots W_{x/2-1/2,x/2} W^*_{x/2,x/2+1/2}\cdots W^*_{x-1/2,x}$ is a unitary matrix acting on $(\mathbb{C}^d)^{\otimes(2x+1)}$ (the "double gate" $W_{x,x+1/2} = W$ is defined in (27)). We then have

$$|\langle\psi|\mathcal{V}_x[\mathbb{1}]|\psi\rangle| = |\langle\circ\psi|\mathfrak{U}\mathfrak{S}|\circ\psi\rangle| \leq |\langle\psi|\psi\rangle|, \qquad (93)$$

where $\mathfrak{S}$ is the periodic shift by one site in $(\mathbb{C}^d)^{\otimes(2x+1)}$ ($\mathfrak{S} = P_{x,0}P_{0,1/2}\cdots P_{x-1/2,x}$ where $P_{i,j}$ it the elementary transposition). In the last step we used that $\langle\circ|\circ\rangle = 1$ and both $\mathfrak{S}$ and $\mathfrak{U}$ are unitary.

## B  Local Conservation Laws in dual-unitary Circuits

In this appendix we study how the continuity equations (53) and (54) simplify in the case of dual-unitary circuits, proving explicitly Eqs. (55) and (56). Specifically, since the manipulations are completely analogous, we only show how to go from (53) to (55). Tracing on the first two sites (i.e. contracting with two bullets $\circ\circ$ from above) we have that the l.h.s. of (53) is 0 by dual unitarity: we thus find

$$J_x^- = q_x^- + J_x^{-\prime\prime}, \qquad (94)$$

where we introduced

$$J_x^{-\prime\prime} \equiv \text{tr}_{x-1,x-\frac{1}{2}}[J_{x-1}^-]. \qquad (95)$$

Tracing again on the first two sites of (94) we have

$$J_{x+1}^{-\prime\prime} = \text{tr}_{x,x+\frac{1}{2}}[J_x^{-\prime\prime}]. \qquad (96)$$

This equation has, as unique solution

$$J_x^{-\prime\prime} = \underbrace{\mathbb{1}\otimes\cdots\otimes\mathbb{1}}_{r-1}, \qquad (97)$$

as can be proven by expanding in an Hilbert-Schmidt orthogonal basis

$$\underbrace{a^{\alpha_1}\otimes\cdots\otimes a^{\alpha_{r-1}}}_{r-1}, \qquad \alpha_j = 1,\ldots,d^2, \qquad (98)$$

where $a^0 = \mathbb{1}$. Plugging (97) back into (94) we finally find

$$J_x^- = q_x^- + \underbrace{\mathbb{1}\otimes\cdots\otimes\mathbb{1}}_{r+1}, \qquad (99)$$

which gives directly (55).

## C  Definition of the Gate

Following Ref. [56], a general dual-unitary gate can be (up to a gauge transformation) parametrised as

$$U = \exp\left[-i\left(\frac{\pi}{4}\sigma_1\otimes\sigma_1 + \frac{\pi}{4}\sigma_2\otimes\sigma_2 + J\sigma_3\otimes\sigma_3\right)\right]\cdot(v_-\otimes v_+), \qquad (100)$$

where $\phi, J \in \mathbb{R}$, $v_\pm \in \mathrm{SU}(2)$. In all numerical computations reported in this paper we take the ultralocal unitaries equal and parametrise them as

$$v_- = v_+ = \begin{pmatrix} re^{i\phi/2} & -\sqrt{1-r^2}e^{-i\theta/2} \\ \sqrt{1-r^2}e^{i\theta/2} & re^{-i\phi/2} \end{pmatrix}, \qquad r \in [0,1], \qquad \theta, \phi \in [0, 4\pi]. \quad (101)$$

# D  All Models Obeying Eq. (86) are Gauge Equivalent to the Self-Dual Kicked Ising Model

Here we show that all circuits with local gates fulfilling (86) are gauge equivalent to self-dual kicked Ising model. From the first condition in Eqs. (86) it follows:

$$V[J]^\dagger (u_+^\dagger \alpha_+ u_+ \otimes \mathbb{1}) V[J] = \tilde{w} \otimes v_+ \alpha v_+^\dagger, \quad (102)$$

where $\tilde{w}$ is some $SU(2)$ matrix and we used the standard parametrisation of the gate used in [56]. These conditions can only be satisfied for $J \in \{\frac{\pi}{4}, 0\}$.

Demanding $w \neq \mathbb{1}$ (the case $w = \mathbb{1}$ is treated in [60]) leads to the solution $J = 0$ and $V[0]^\dagger (\sigma_1 \otimes \mathbb{1}) V[0] = \sigma_3 \otimes \sigma_2$ or $V[0]^\dagger (\sigma_2 \otimes \mathbb{1}) V[0] = -\sigma_3 \otimes \sigma_1$. Proceeding with the first condition we have:

$$u_+^\dagger v_+^\dagger \sigma_2 v_+ u_+ = \sigma_1, \qquad \alpha_+ = v_+ \sigma_2 v_+^\dagger. \quad (103)$$

The analogue conditions for the unknowns with a minus sign are derived in the same manner. Using gauge transformation (13), we may set $\alpha_\pm = \sigma_3$ in order to have the additional eigenvectors of the form (87). The equation (103) is solved by:

$$v_\pm u_\pm = e^{i\psi_\pm} \begin{pmatrix} r_\pm & i\sqrt{1-r_\pm^2} \\ \sqrt{1-r_\pm^2} & -ir_\pm \end{pmatrix}, \quad (104)$$

which generate a two-parameter $r_\pm$ family of models (the phase $\psi_\pm$ is irrelevant). The reflection symmetric case $r_+ = r_- = \cos h$ is therefore gauge equivalent to the self-dual kicked Ising model, with $h$ being the magnetic field in the $z$ direction. This is also seen in the eigenvalues of the maps $\mathcal{M}_{\pm,U}$ (cf. (77)), which exactly match.

# E  Numerical methods

Calculating the operator entanglement entropy numerically is computationally expensive with resources scaling exponentially with $t$. In our case, we iteratively constructed the corner transfer matrix $\mathcal{C}[a]$, as defined in (38). First we construct the doubled gate $W$, from which we build the first row of $\mathcal{C}[a]$. Then we add additional precomputed rows via matrix computations until we end up with the final corner transfer matrix. In the last and by far the most expensive step we calculate $d^{2(x_+ + x_-)}$ matrix elements, with each costing $d^{2x_+}$ operations. At $y = 0$, $d = 2$ the total cost scales as $2^{6t}$, which is still much better than using the row/column transfer matrices $\mathcal{H}_x[a]$ and $\mathcal{V}_x[a]$, where the cost scales as $2^{8t}$.

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
