# Peer review of "Operator Entanglement in Local Quantum Circuits I: Chaotic Dual-Unitary Circuits"

_SciPost Physics, doi:SciPost Phys. 8, 067 (2020)_

## Round 1 · Referee Report · Jerome Dubail (Referee 1) · 2019-11-6

Strengths

1- Very interesting problem and timely results

2- Beautiful “dual unitary” models solvable by new methods; new definition of “maximal chaos” in that context

3- Exact results

4- Interesting new conjecture about growth of operator entanglement, strongly supported by numerics

Weaknesses

1- Only minor ones. The physical discussion of the conjecture and its possible implications could probably be expanded. A few technical steps in the derivation of the results could be made more accessible to the reader.

Report

This is an excellent paper with very interesting and timely results on a difficult problem. The authors provide a detailed study of the growth of operator entanglement of local operators in Heisenberg picture (a quantity they propose to call 'local operator entanglement') in a class of solvable one-dimensional chaotic quantum spin chains. Those 'dual-unitary' models were introduced by the authors themselves in a series of recent papers.

The results presented here provide analytical support for claims made in other recent papers on local operator entanglement: this quantity grows linearly with time in chaotic spin chains. This is in contrast with non-chaotic systems, where it is conjectured to grow at most logarithmically.
Also, an interesting new conjecture is presented about the exact rate of the linear growth, which is well supported by numerical results.

I recommend publication of this manuscript in Scipost. The authors may want to consider the suggestions/remarks below before publication.

Requested changes

1- Abstract: perhaps it should be said explicitly that 'maximally-chaotic' here refers to a specific definition $-$valid in the context of dual-unitary models$-$ which is new and appears in this paper for the first time.

2- Eq. (2): clearly in the product $U^{\otimes L}$ each $U$ is supposed to act on two neighboring sites; yet this should be written explicitly

3- Graphical representation of Eq. (12): maybe the authors could put the site indices ($\dots, -1, -\frac{1}{2},0, \frac{1}{2}, 1 ,\dots $), as they do in Eq. (10), on the top and bottom rows. I believe this would make the conventions clearer. Also, maybe it would be clearer to draw all the sites, from $-\frac{L}{2}$ to $\frac{L}{2}$, just for this picture, so that it's clear that $a$ is inserted at $x=0$.

4- I find that it is hard to visualize the operations that bring us from Eq. (29) to Eq. (32). Since Eq. (32) plays an important role in the rest of the paper, its derivation would deserve to be slightly expanded. Perhaps having both Eqs. (29) and (32) drawn with the same small values of $y$ and $t$ would help. And perhaps a picture of the intermediate step, where Eq. (26b) is used, would help as well.

5- before Eq. (47): 'any such transfer matrix is a contracting operator': why is that so? Is it obvious?

6- Eqs. (49)-(50). If the conventions are that the operators act from bottom to top in the drawings, then the rainbow state in Eq. (49) should appear with open legs on top. More importantly, there may be an error in Eq. (50): if I am not mistaken, the conventions are such that the squared norms $\left< \cup \left. \right| \cup \right>$ and $\left< \circ \circ \left. \right| \circ \circ \right>$ are both equal to $1$, and then this implies $\left< \cup \left. \right| \circ \circ \right> = \frac{1}{d}$. If so, then $d$ should be replaced by $1/d$ in Eq. (50), in order for $\left| \bar{r}_l \right>$ to be orthogonal to $\left| r_l \right>$.

7- Around Eq. (51): perhaps it could be recalled that $\left| \bar{r}{x+} \right>$ and $\left| e_{x_+} \right>$ are the same state. More importantly, the identity $\left< e_{x_+} \left. \right| a^\dagger \circ \dots \circ a \right>=d^{1-x_+}/\sqrt{d^2-1}$ is sufficiently non-trivial so that its derivation would deserve to be expanded a bit. In particular, isn't that result relying on the assumption ${\rm tr}[a] = 0$? (If so, this assumption needs to be stated explicitly)

8- I'm confused by Eqs. (61) and (62): looking at the graphical Eq. (59), wouldn't one expect $\mathcal{M}_+$ to appear on the left of $\mathcal{M}_-$ in Eqs. (61)-(62)?

9- Fig. 2 and discussion below Eq. (65): the discussion of the von Neumann entropy could probably be expanded there. First, since the results of the paper rely on $n$ integer $\geq 2$, where do the claims about von Neumann come from exactly? Second, it is very interesting that the von Neumann entropy always has maximal growth, in contrast with the Renyi entropies; this seems to be a very non-trivial observation which deserves to be highlighted.

10- section 5 and the new conjecture: the conjecture is well described mathematically, and the numerical evidence supporting it is well presented. However, I find that its physical meaning and implications are not clearly discussed. For instance, if the growth is linear then the local operator entanglement $S(y,t)$ is expected to behave as $t f(y/t)$ for some function $f$ at large $t$ and fixed ratio $y/t$. It seems that the conjecture Eq. (66) will typically lead to an interesting profile $f(v) = {\rm min}[ s_- (1-v), s_+ (1+v) ]$, for two constants $s_-$ and $s_+$ given by Eqs. (55) and (65), and for $s_- \neq s_+$ this profile will be asymmetric and will look different from the symmetric 'pyramid' found in Ref. [37] (see Fig. 13 in that ref.). Will this simply come from the absence of reflection symmetry of the dual unitary model? This is something that could be discussed. More generally I think it would be interesting for some readers to have a discussion of the physical meaning and potential consequences of the new conjecture.

---

## Round 1 · Referee Report · Anonymous (Referee 2) · 2020-2-10

Strengths

1: Relevant problem: characterization of quantum chaotic behavior by entanglement of local operators. 2: Precise statements of goals, techniques and results 3: Interesting models proposed, with a clear algebraic formulation allowing for exact limit results and conjecture on asymptotic behavior backed by numerical datas.

Weaknesses

1: Technical aspects should be precised 2: Link between two definitions of maximally chaotic should be established 3: Conjecture on asymptotic behavior as combination of limit behaviors should be justified a priori, or at least interpreted.

Report

A very interesting, very clear and well written paper on a highly relevant subject which has attracted much attention recently. Many interesting analytical results, and a highly non trivial conjecture backed by excellent numerical datas. Recommanded for publication after some issues are clarified.

Requested changes

1: "Maximally chaotic" is characterized in the Introduction as "no conservation law" (which seems perfectly relevant in an integrability context) whereas it is characterized in Def. 4.1 as "only (48) are eigenvectors of [either row transfer matrix] with eigenvalues of unit magnitude". How does one go from one definition to the other ?

2: A related question would be: how to prove that given "dual unitary quantum circuits" have NO dynamical conservation laws ? Already not obvious to me regarding local conservation laws; Paper II shows that specific dual quantum circuits may exhibit local conservation laws; how to prove their absence is more complicated. Even worse: possible non-local conservation laws are usually quite difficult to construct, hence their absence is probably quite difficult to prove either.

3: Why are transfer matrices contracting operators (eqn. 47) ?

4: The two statements after Definition 4.1 seem contradictory: "... (48) are generically only a subset of the eigenvectors of [...] associated with eigenvalue 1" vs. "For generic dual unitary circuits there are no eigenvectors of [...] with unit magnitude eigenvalue other than (48)". The meaning of "generic" in both cases should probably be clarified.

5: What would be the meaning of the conjecture 5.1 ? Is there a physical interpretation of this decoupling of asymptotic behaviour at large x,t as a sum of two non-interacting limits : (x-) to infinity + (x+) to infinity ?

---

## Round 2 · Referee Report · Jerome Dubail (Referee 1) · 2020-3-15

Report

The authors answered all my comments in a very satisfactory way. I am happy to recommend this very nice work for publication.

---

## Round 2 · Referee Report · Anonymous (Referee 2) · 2020-3-16

Report

Questions of this referee suitably and clearly answered. I accept this paper for publication.

---

## Round 2 · Author Response

Dear Editor,

We thank the two referees for their careful reading of our paper, for their positive assessment, and for their many useful suggestions. Here we provide a point-by-point reply to their points.

Reply to Referee 1 (Dr Dubail)

"1- Abstract: perhaps it should be said explicitly that 'maximally-chaotic' here refers to a specific definition − valid in the context of dual-unitary models − which is new and appears in this paper for the first time.
"

Response: We agree. We modified the relevant sentence in the abstract as follows:

We identify a class of “completely chaotic” dual-unitary circuits where the local-operator entanglement grows linearly and we provide a conjecture for its asymptotic behaviour which is in excellent agreement with the numerical results.”
Note that we changed the name of our special class from “maximally chaotic” to “completely chaotic” (see the response to the first point of Referee 2).

"2- Eq. (2): clearly in the product U⊗L each U is supposed to act on two neighboring sites; yet this should be written explicitly"

Response: We clarified the equation.

"
3- Graphical representation of Eq. (12): maybe the authors could put the site indices (…,−1,−1/2,0,1/2,1,…), as they do in Eq. (10), on the top and bottom rows. I believe this would make the conventions clearer. Also, maybe it would be clearer to draw all the sites, from −L/2 to L/2, just for this picture, so that it's clear that a is inserted at x=0."

Response: We put the site indices in the bottom row of Eq. 12. We decided not to follow the second suggestion: since the operator is inserted at a generic y we prefer to use a generic length L instead of specifying one.

"
4- I find that it is hard to visualize the operations that bring us from Eq. (29) to Eq. (32). Since Eq. (32) plays an important role in the rest of the paper, its derivation would deserve to be slightly expanded. Perhaps having both Eqs. (29) and (32) drawn with the same small values of y and t would help. And perhaps a picture of the intermediate step, where Eq. (26b) is used, would help as well."

Response: We added the intermediate step.

"5- before Eq. (47): 'any such transfer matrix is a contracting operator': why is that so? Is it obvious?
"

We added a proof in the new Appendix A.

"
6- Eqs. (49)-(50). If the conventions are that the operators act from bottom to top in the drawings, then the rainbow state in Eq. (49) should appear with open legs on top. More importantly, there may be an error in Eq. (50): if I am not mistaken, the conventions are such that the squared norms ⟨∪|∪⟩ and ⟨∘∘|∘∘⟩ are both equal to 1, and then this implies ⟨∪|∘∘⟩=1/d. If so, then d should be replaced by 1/d in Eq. (50), in order for |\bar r_\ell> to be orthogonal to |r_\ell>."

Response: We agree with the first observation and we accordingly flipped the rainbow state. Concerning the second point, there probably was a misunderstanding: we do not require \bar r_ell to be orthogonal to r_ell, we want bar r_\ell for different \ell to be orthogonal. The states defined in Eq. 51 (50 in the old version) have this property. 

"7- Around Eq. (51): perhaps it could be recalled that  |\bar r_{x_+}>  and |e_{x_+}⟩ are the same state. More importantly, the identity ⟨ex+|a†∘⋯∘a⟩=d1−x+/√d2−1 is sufficiently non-trivial so that its derivation would deserve to be expanded a bit. In particular, isn't that result relying on the assumption tr[a]=0? (If so, this assumption needs to be stated explicitly)"

Response: We added the additional explanations requested. 

"8- I'm confused by Eqs. (61) and (62): looking at the graphical Eq. (59), wouldn't one expect M+ to appear on the left of M− in Eqs. (61)-(62)?"

Response: Equations 79 and 80 (of the new version) were actually correct although counter-intuitive. There was a problem in Eq. 78 (of the new version). We fixed these issues and hope that this step is now clearer.

"

9- Fig. 2 and discussion below Eq. (65): the discussion of the von Neumann entropy could probably be expanded there. First, since the results of the paper rely on n integer ≥ 2, where do the claims about von Neumann come from exactly? Second, it is very interesting that the von Neumann entropy always has maximal growth, in contrast with the Renyi entropies; this seems to be a very non-trivial observation which deserves to be highlighted."

Response: We thank Dr Dubail for highlighting this point. First, as it is now stressed in the text, the von Neumann entropy is computed by performing an analytical continuation in n of Eq. 65 and then taking the limit n->1^+. Second, this point is now mentioned in the beginning of Sec. 5 and the Conclusions.

"

10- section 5 and the new conjecture: the conjecture is well described mathematically, and the numerical evidence supporting it is well presented. However, I find that its physical meaning and implications are not clearly discussed. For instance, if the growth is linear then the local operator entanglement S(y,t) is expected to behave as t f(y/t) for some function f at large t and fixed ratio y/t. It seems that the conjecture Eq. (66) will typically lead to an interesting profile f(v)=min[s−(1−v),s+(1+v)], for two constants s− and s+ given by Eqs. (55) and (65), and for s−≠s+ this profile will be asymmetric and will look different from the symmetric 'pyramid' found in Ref. [37] (see Fig. 13 in that ref.). Will this simply come from the absence of reflection symmetry of the dual unitary model? This is something that could be discussed. More generally I think it would be interesting for some readers to have a discussion of the physical meaning and potential consequences of the new conjecture."

Response: we expanded the discussion of the conjecture following these and the suggestions of Referee B. In particular the expanded discussion contains both the main points mentioned by Dr Dubail.

Reply to Referee 2

"1: "Maximally chaotic" is characterized in the Introduction as "no conservation law" (which seems perfectly relevant in an integrability context) whereas it is characterized in Def. 4.1 as "only (48) are eigenvectors of [either row transfer matrix] with eigenvalues of unit magnitude". How does one go from one definition to the other ?"

Thanks for noting this inconsistency. The two definitions are indeed not equivalent, in particular that of Def. 4.1 is more stringent than the one given in the introduction. Moreover, the name maximally chaotic could be confusing as only a subset of these systems have operator entanglement growing at the maximal speed. To solve this issue

i) We removed the adverb “maximally”.

ii) We stated in the introduction that chaotic systems are systems with no local conservation laws

iii) We called “completely chaotic” the circuits of Def 4.1.

After the definition we also discuss the relation between chaotic and completely chaotic dual unitary circuits and show that the property of being “completely chaotic" is more stringent than that of being chaotic (i.e. having no local conservation laws).

"

2: A related question would be: how to prove that given "dual unitary quantum circuits" have NO dynamical conservation laws ? Already not obvious to me regarding local conservation laws; Paper II shows that specific dual quantum circuits may exhibit local conservation laws; how to prove their absence is more complicated. Even worse: possible non-local conservation laws are usually quite difficult to construct, hence their absence is probably quite difficult to prove either.
"

We completely agree with the referee. In the new discussion following Def. 4.1 we focus on local conservation laws (written as sums of local densities) and show what happens if non-trivial local conservation laws are combined with dual-unitarity.

"
3: Why are transfer matrices contracting operators (eqn. 47) ?"

A proof is added in the new Appendix A.

"
4: The two statements after Definition 4.1 seem contradictory: “... (48) are generically only a subset of the eigenvectors of [...] associated with eigenvalue 1" vs. "For generic dual unitary circuits there are no eigenvectors of [...] with unit magnitude eigenvalue other than (48)”. The meaning of "generic" in both cases should probably be clarified."

We meant that completely chaotic circuits are a subclass, but are the most common case. In other words if one generates a dual-unitary gate at random (e.g. using Eq. 23 of Ref. 55 with random u_- , u_+, v_- , v_+) it is completely chaotic with probability one (for all J \neq \pi/4). We modified the paper to clarify this point.

"5: What would be the meaning of the conjecture 5.1? Is there a physical interpretation of this decoupling of asymptotic behaviour at large x,t as a sum of two non-interacting limits : (x-) to infinity + (x+) to infinity ?"

We added an expanded discussion of the conjecture following this suggestion and that of Dr Dubail.

---

## Round 2 · List of Changes

1- The name "maximally chaotic" has been replaced by "completely chaotic" throughout the paper. We also removed the adverb "maximally" from the title.

2- A sentence in the abstract has been changed

3- Site labels added to eq 12

4- Discussion after eq 28 improved

5- Eq 32 added

6- Discussion after Definition 4.1 expanded (including new equations 52-65)

7- Improved discussion in Sec 5 (including new equation 68)

8- Equations 78 corrected

9- Improved discussion in Sec 5.2 (including new equation 86)

10- Improved discussion in the conclusions.

11 - New Appendices A and B added

---

## Editorial Decision

published